# Evidence Regularization for Multimodal Deep Evidential Regression

**Zhimin Shao**[1]                                                         SZM21@MAILS.TSINGHUA.EDU.CN
[1] *Department of Electronic Engineering, Tsinghua University, Beijing, China*

**Weibei Dou**[1]                                                         DOUWB@TSINGHUA.EDU.CN

## Abstract

Uncertainty estimation is crucial in cost-sensitive areas, especially in the medical field, where multimodal information is common and effective. Existing studies have found the zero-confidence issue in unimodal settings, while the analysis in multimodal scenarios is lacking. In this work, we introduce the confidence paradox, where unimodal uncertainty is high but decreases after fusion, and present evidence of regularization to tackle this issue. Initial results on the cubic and CT slice datasets show reduced root mean squared errors and improved detection of out-of-distribution samples, improving predictive reliability and training stability.

**Keywords:** Uncertainty estimation, Multimodal, Deep learning, Regression

## 1. Introduction

In medical image analysis, regression tasks predict continuous values $y$ in $\mathbb{R}^d$ using input images or features $X$, crucial for applications like forensic age estimation (Halabi et al., 2019), disease score regression (Stonnington et al., 2010), and image segmentation by identifying boundaries (Yin et al., 2020). With the increasing prevalence of multimodal data in the medical field and the necessity for reliable predictions, assessing the predictive uncertainty in deep learning models for each data modality and the combined uncertainty following multimodal data integration is of paramount importance (Ma et al., 2021).

Predictive uncertainty is divided into aleatoric (data variability) and epistemic (model knowledge gaps). While Bayesian methods like dropout (Gal and Ghahramani, 2016) and deep ensemble (Lakshminarayanan et al., 2017), infer the posterior distribution of parameters or observed targets to estimate uncertainty, challenges like computational intensity and disentanglement between these uncertainty types. Deep evidential regression (DER) (Amini et al., 2020) and its multimodal version (Ma et al., 2021) have revolutionized uncertainty estimation by probabilistically estimating target distribution parameters and enhancing the separation of uncertainties but faced issues like zero-confidence in training. Specific activation functions were employed to ensure non-negative evidential distribution parameters, which limits the models' learning efficiency from training data (Pandey and Yu, 2023). To bypass the need for selecting specific activation functions, dual activation function fusion was introduced (Shao et al., 2024), yet the zero-confidence issue persists. The issue, identified in classification (Pandey and Yu, 2023) and regression (Ye et al., 2024), leads to the cessation of model updates in zero-confidence areas during training. Although evidence regularization has successfully mitigated the zero-confidence issue in unimodal settings (Pandey and Yu, 2023; Ye et al., 2024), multimodal regression's complexities necessitate extending this approach to improve training stability and uncertainty management.

## 2. Confidence Paradox in Multimodal Scenario

Deep evidential regression (Amini et al., 2020) assumes that a target value $y$ is i.i.d from a Gaussian distribution $\mathcal{N}(\mu, \sigma^2)$. The unknown mean $\mu$ and variance $\sigma$ are presumed to follow a Normal Inverse-Gamma (NIG) distribution $(\mu, \sigma^2) \sim NIG(\gamma, v, \alpha, \beta)$, where $\mu \sim \mathcal{N}(\gamma, \frac{\sigma^2}{v})$ and $\sigma^2 \sim \Gamma^{-1}(\alpha, \beta)$. $\Gamma(\cdot)$ is the gamma function. The parameters of NIG $\boldsymbol{m} = (\gamma, \nu, \alpha, \beta)$ are determined by the neural network's output $\boldsymbol{o} = (o_\gamma, o_\nu, o_\alpha, o_\beta) = f(X|\boldsymbol{\theta})$ with $\boldsymbol{\theta}$ as the network's trainable parameters. To enforce the constraints on $\boldsymbol{m}$, a SoftPlus activation function is used to generate $(\nu, \alpha, \beta)$, with an additional increment of 1 for $\alpha$, and a linear activation function for $\gamma \in \mathbb{R}$. For multimodal scenarios (Ma et al., 2021), $y \sim \sum_{m=1}^{M} \frac{1}{M} NIG(\gamma_m, \nu_m, \alpha_m, \beta_m)$. The parameters $(\gamma_m, \nu_m, \alpha_m, \beta_m)$ are learned from training data in $m$-th modality by $f_m(\cdot)$. The summation of NIG distributions is

$$NIG(\gamma, \nu, \alpha, \beta) \triangleq NIG\left(\gamma_1, \nu_1, \alpha_1, \beta_1\right) \oplus NIG\left(\gamma_2, \nu_2, \alpha_2, \beta_2\right) \oplus \cdots \oplus NIG\left(\gamma_M, \nu_M, \alpha_M, \beta_M\right) \tag{1}$$

where the summation $\oplus$ for any two NIG distributions is defined as

$$\begin{aligned} \gamma &= (\nu_1 + \nu_2)^{-1}(\nu_1\gamma_1 + \nu_2\gamma_2), \quad \alpha = \alpha_1 + \alpha_2 + \tfrac{1}{2} \\ \nu &= \nu_1 + \nu_2, \quad \beta = \beta_1 + \beta_2 + \tfrac{1}{2}\nu_1\left(\gamma_1 - \gamma\right)^2 + \tfrac{1}{2}\nu_2\left(\gamma_2 - \gamma\right)^2 \end{aligned} \tag{2}$$

Based on NIG distribution, we use $\mathbb{E}[\mu] = \gamma$ as the prediction, $\mathbb{E}\left[\sigma^2\right] = \frac{\beta}{(\alpha-1)}$ as the aleatoric uncertainty, and $\mathrm{Var}[\mu] = \frac{\beta}{v(\alpha-1)}$ as the epistemic uncertainty.

In the zero-confidence area, characterized by the parameter $\alpha = 1$, both the aleatoric and epistemic uncertainties diverge to infinity due to the term $\alpha - 1 = 0$ in the denominator. Consequently, $\mathrm{SoftPlus}(o_\alpha) = 0$ and $\frac{\partial \alpha}{\partial o_\alpha} = 0$. Therefore, the gradient of loss $\frac{\partial \mathcal{L}}{\partial o_\alpha} = \frac{\partial \mathcal{L}}{\partial \alpha}\frac{\partial \alpha}{\partial o_\alpha}$ is also zero. The model would stick in the zero-confidence area once it falls into it, leading to infinite values for both aleatoric and epistemic uncertainties in unimodality. However, this scenario presents a paradox where the aggregation of two highly uncertain NIG distributions results in a significant reduction of the overall uncertainty, which contradicts intuition, shown in Table 1.

Commonly, we would anticipate that incorporating more comprehensive multimodal data would reduce uncertainty. However, deriving a certain decision based on completely uncertain evidence from each modality is illogical and highlights a critical flaw in the uncertainty modeling approach. Traditional regularization on predictive-level fused $\alpha$ cannot propagate to every sub-modality.

Table 1: An Numeric Example of Confidence Paradox

|  | NIG$(\gamma, \nu, \alpha, \beta)$ | AU | EU |
|---|---|---|---|
| Mod 1 | NIG(0, 1, 1, 1) | Inf. | Inf. |
| Mod 2 | NIG(1, 1, 1, 1) | Inf. | Inf. |
| Sum | NIG(0.5, 2, 2.5, 2.25) | 1.5 | 0.75 |

## 3. Evidence Regularization for Multimodal DER (ER-MDER)

Similar to (Ma et al., 2021), the overall loss function for multimodal learning is the sum of losses of multiple modalities ($m$ denotes the $m$-th modality), the pseudo modality (obtained by features concatenation or concatenation after representation learning, denoted by $P$), and

the predictive-level fused distribution (denoted by $F$), $\mathcal{L} = \sum_{m=1}^{M} \mathcal{L}_m(\mathrm{w}) + \mathcal{L}_P(\mathrm{w}) + \mathcal{L}_F(\mathrm{w})$. To avoid the confidence paradox, we put constraints directly on $o_\alpha$ for each modality.

$$\mathcal{L}(\mathrm{w}) = \mathcal{L}^{\mathrm{NLL}}(\mathrm{w}) + \lambda_1 \mathcal{L}^{\mathrm{R}}(\mathrm{w}) + \lambda_2 \mathcal{L}^{\mathrm{O}}(\mathrm{w})$$

$$= \frac{1}{2}\log\left(\frac{\pi}{v}\right) - \alpha \log(\Omega) + \left(\alpha + \frac{1}{2}\right)\log\left((y_i - \gamma)^2 v + \Omega\right) + \log\left(\frac{\Gamma(\alpha)}{\Gamma\left(\alpha + \frac{1}{2}\right)}\right) \quad (3)$$

$$+ \lambda_1\left(|y - \gamma| \cdot (2\nu + \alpha)\right) - \lambda_2\left(|y - \gamma| \cdot o_\alpha\right)$$

where $\Omega = 2\beta(1 + \nu)$. While there is no $o_\alpha$ for the fused distribution, $\mathcal{L}_F(\mathrm{w})$ only contains two items $\mathcal{L}^{\mathrm{NLL}}(\mathrm{w}) + \lambda_1 \mathcal{L}^{\mathrm{R}}(\mathrm{w})$.

## 4. Experiments

**Cubic Regression** We evaluate our proposed ER-MDER in comparison with the multimodal DER (MDER) on the cubic regression dataset, particularly within zero-confidence areas. Following (Amini et al., 2020), models were trained on $y = x^3 + \epsilon$ with $\epsilon \sim \mathcal{N}(0, 3)$ over the interval $x \in [-4, 4]$ and tested in the range $x \in [-6, -4) \cup (4, 6]$. The $m$-th modality input $x_m$ equals to $x + \epsilon_x, \epsilon_x \sim \mathcal{N}(0, 0.01)$. After evidential regularization, our model can update faster than the original one within the zero-confidence area and generate more reliable predictions, shown in Figure 1.

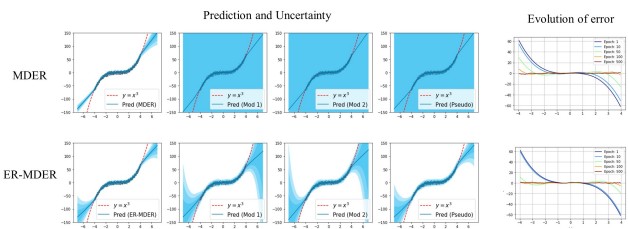

Figure 1: Prediction, uncertainty estimation and error evolution in MDER and ER-MDER. The blue shade represents prediction uncertainty.

**CT Slices Location** We evaluated our methods on CT slices dataset (Graf et al., 2011). Each image includes two modalities: one with 240 attributes for bone structure and another with 144 attributes for air inclusions. The objective is to predict the image's axial axis position, ranging from 0 to 180. Similar settings were used as (Ma et al., 2021). We introduce noise ($\epsilon = 0.1$) to half of the test samples to create out-of-distribution (OOD) samples and distinguish them using uncertainty. According to Table 2, our method achieves lower Root Mean Squared Error (RMSE) and enhanced performance in OOD detection, as indicated by a higher AUROC score.

Table 2: Results of prediction and OOD detection in CT Slices.

|  | RMSE($\downarrow$) | AUROC($\uparrow$) |
|---|---|---|
| MDER | 0.79 | 0.615 |
| ER-MDER (Ours) | 0.67 | 0.956 |

Future work will focus on more experiments on public medical image datasets, and application to in-house data.

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
