# OpenReview forum: "Evidence Regularization for Multimodal Deep Evidential Regression"
_MIDL.io/2024/Short_Papers — MIDL 2024 Short Papers_

### Official Review · Reviewer_t4Be · 2024-04-25

**Confidence:** 4
**Final Rating:** 4

**Review:**

This study focuses on the regularisation of evidence to improve the quality of uncertainty estimates in a multimodal scenario. Results shows the interest of the proposed regularization scheme on both synthetic and real CT images.

The strengths of this work are:
1) the relevance of the theme of this work, any improvement/innovation of which could have a major impact in our field
2) the relevance of the experiments and the metrics that have been chosen
3) the quality of the results

The main weakness of this article concerns the didactic aspect. The theme of uncertainty is not necessarily easy to understand for the uninitiated, and strong educational efforts on key points of the article should be strengthened. It is important to explain the signification or the influence of each of the parameters of the Normal Inverse Gamma distribution, which is key in this study. For instance, it is stated that the parameter alpha should be equal to one in the zero-confidence area without any explanation, making the understanding of the article complicated.

---

### Decision · Program_Chairs · 2024-04-26

Accept